# Beyond Next Token Prediction: Patch-Level Training for Large Language Models

**Chenze Shao, Fandong Meng**[∗]**, Jie Zhou**
Pattern Recognition Center, WeChat AI, Tencent Inc, China
{`chenzeshao,fandongmeng,withtomzhou`}`@tencent.com`

## Abstract

The prohibitive training costs of Large Language Models (LLMs) have emerged as a significant bottleneck in the development of next-generation LLMs. In this paper, we show that it is possible to significantly reduce the training costs of LLMs without sacrificing their performance. Specifically, we introduce patch-level training for LLMs, in which multiple tokens are aggregated into a unit of higher information density, referred to as a 'patch', to serve as the fundamental text unit for training LLMs. During patch-level training, we feed the language model shorter sequences of patches and train it to predict the next patch, thereby processing the majority of the training data at a significantly reduced cost. Following this, the model continues token-level training on the remaining training data to align with the inference mode. Experiments on a diverse range of models (370M-2.7B parameters) demonstrate that patch-level training can reduce the overall training costs to 0.5×, without compromising the model performance compared to token-level training. Source code: `https://github.com/shaochenze/PatchTrain`.

## 1 Introduction

Large Language Models (LLMs, Achiam et al., 2023; Touvron et al., 2023a;b; Team et al., 2023; Bai et al., 2023) have achieved remarkable progress in language understanding and generation, which are primarily attributed to their unprecedented model capacity and the corresponding growth in the volume of training data they require (Kaplan et al., 2020; Hoffmann et al., 2022). However, this scaling up comes with a substantial rise in computational costs, making the training efficiency of LLMs a critical concern. Despite the ongoing efforts on efficient LLMs (Wan et al., 2023), it remains a formidable challenge to reduce training costs without compromising the model performance.

Specifically, the amount of compute (FLOPs) required for training LLMs is approximately proportional to both the number of model parameters $N$ and the number of text units (i.e., tokens) $D$ in the training data. This relationship can be expressed as:

$$C \approx 6ND. \tag{1}$$

Therefore, strategies for reducing training costs can target either the reduction in the number of model parameters $N$ or the number of text units $D$. One prominent approach to reduce the parameter size $N$ is called model growth (Gong et al., 2019; Yang et al., 2020; Chen et al., 2022). Rather than equipping the model with full parameters from the beginning, it advocates for a progressive expansion of the model's parameter size throughout the training phase, thereby reducing the average parameter size during training. Nonetheless, a model's performance hinges on an adequate number of parameters to store extensive knowledge and develop intricate reasoning capabilities. The inadequacy of parameters inherently limits the scope of knowledge and capabilities a model can develop, rendering it challenging to match the performance of training with the full parameter set.

The second pathway to lowering training costs is reducing the number of text units $D$ within the training data. This direction remains largely unexplored but intuitively holds more promise, as the knowledge embedded within training data is sparsely distributed across numerous tokens, with each token encapsulating a minimal amount of information. This sparse distribution of information results

---

[∗]Corresponding author.

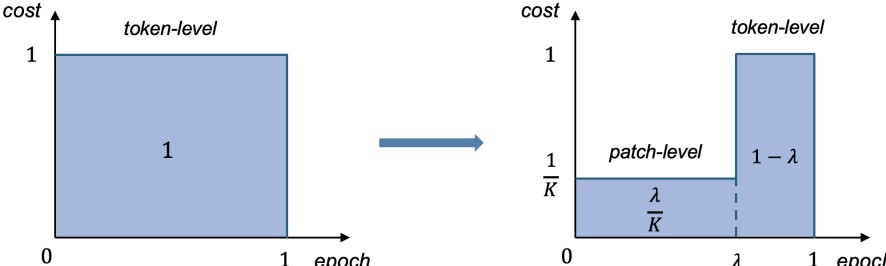

Figure 1: Visualization of overall training costs with patch compression for a fraction $\lambda$ of training data and patch size $K$.

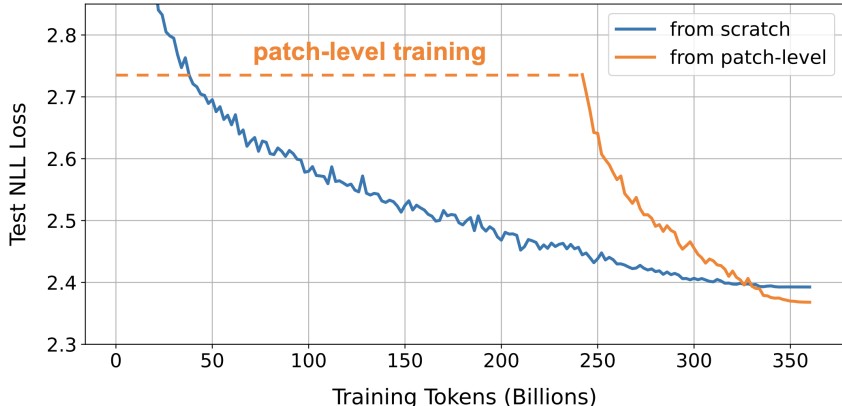

Figure 2: Negative log-likelihood (NLL) loss on test set w.r.t the number of processed tokens during the training of 370M-parameter Transformers.

in a scenario where, despite the substantial computational costs incurred during each learning step, only a small fraction of model parameters that are relevant to the current token are effectively updated. By increasing the amount of information the model processes at each learning step—that is, by augmenting the information density of text units and thus reducing the number of text units $D$—we could potentially boost training efficiency significantly without sacrificing model performance.

Building on these insights, this paper introduces patch-level training for large language models, in which multiple tokens are aggregated into a unit of higher information density, referred to as a 'patch', to serve as the fundamental text unit for training LLMs. Specifically, we divide the training process into two stages: patch-level training and token-level training. During patch-level training, we feed the language model shorter sequences of patches and train it to predict the next patch, thereby processing the majority of the training data at a significantly reduced cost. The resulting parameters are used to initialize the token-level model, which then continues training on the remaining data to adapt the knowledge gained during patch-level training to the token-level.

Figure 1 illustrates the efficiency advantage of patch-level training, where the area of the shape represents the overall training costs. With a patch size of $K$, the amount of compute required for patch-level training is $1/K$ of that required for token-level training. When a fraction $\lambda$ of the training data is compressed into patches, the overall training costs are reduced to $\lambda/K + 1 - \lambda$ times the original costs. For instance, to halve the training costs, one could set the patch size $K = 4$ and conduct patch-level training on $\lambda = 2/3$ of the training data.

Employing the above settings ($K = 4, \lambda = 2/3$), we train a series of LLMs of varying sizes (370M-2.7B parameters) on the Pile dataset (Gao et al., 2020). Figure 2 illustrates the trend of NLL loss against the number of training tokens for the 370M model. After initialization with patch-level training, the model experiences a rapid decrease in loss as it continues token-level training on the remaining data. Remarkably, it achieves an even lower loss in comparison with training from scratch, while reducing training costs by half. By further adjusting the hyperparameter settings, even higher acceleration rates can be achieved, with only a slight sacrifice in model performance.

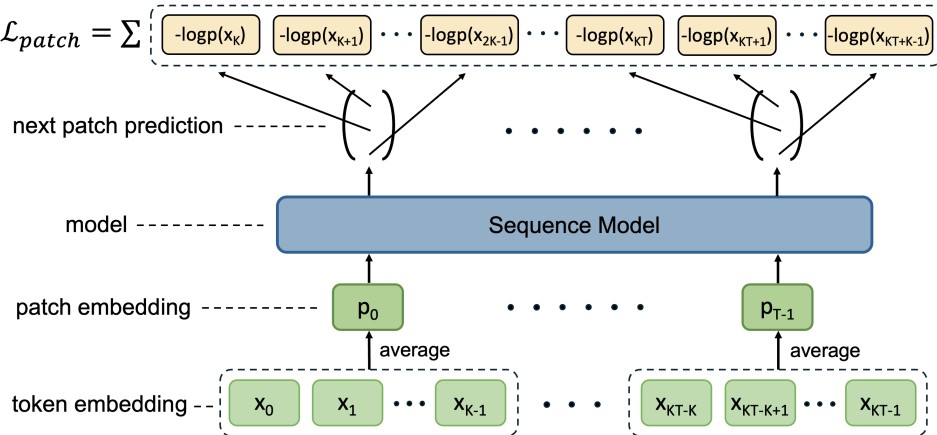

Figure 3: Overview of patch-level training. Every consecutive $K$ token embeddings are averaged to form the patch embedding. The sequence model is fed the patch sequence and trained to predict the next patch. The cross-entropy loss is computed based on each patch prediction vector and all the subsequent $K$ tokens in its next patch.

## 2 PATCH-LEVEL TRAINING

In this section, we outline the patch-level training approach for large language models, as illustrated in Figure 3. Initially, the token sequence is transformed into a patch sequence by compressing every $K$ consecutive tokens into a single patch. This patch sequence is then fed into the sequence model, and the model is trained to predict all tokens in the next patch. The knowledge acquired during patch-level training is subsequently transferred to the token-level model. Specifically, we use the parameters obtained from the patch-level model to initialize the token-level model, and then proceed with token-level training on the remaining data.

While formulating the patch-level model structure, our goal is to minimize the discrepancy between patch-level and token-level models, thereby ensuring that the knowledge gained during patch-level training can be smoothly transferred to the token-level model. In practice, we observed that it is crucial for at least one end (input or output) of the patch-level model to remain consistent with the token-level model, which acts as an anchor to align their representations. The detailed ablation is presented in section 3.7.

Given the context length $T$ for token-level training, we set the context length for patch-level training as $KT$, which is then compressed to a patch sequence of length $T$ to maintain consistency with the subsequent token-level training. To avoid introducing unnecessary parameters during token-to-patch compression, we represent the patch embedding as the average of its associated token embeddings. Let $p_i$ be the $i$-th patch, $x_{iK+k}$ be the $k$-th token in the $i$-th patch, and $E$ be the embedding function. The patch embedding is:

$$E(p_i) = \frac{1}{K} \sum_{k=0}^{K-1} E(x_{iK+k}).$$
(2)

The patch-level model is trained through next patch prediction, i.e., predicting the $K$ tokens in the next patch. The simultaneous prediction of multiple tokens has been explored in speculative decoding, which typically employs multiple output heads and each head is responsible for predicting a distinct token (Cai et al., 2024; Lin et al., 2024). However, this approach would also entail additional parameters that may be unfavorable for the subsequent knowledge transfer. Instead, we maintain a single output head and make its prediction cover all tokens in the next patch. Specifically, we calculate the cross-entropy loss for all the subsequent $K$ tokens based on the same patch prediction $P(\cdot|p_{<i})$, resulting in the following loss function:

$$\mathcal{L}_{patch} = -\sum_{i=1}^{T} \sum_{k=0}^{K-1} \log P(x_{iK+k}|p_{<i}).$$
(3)

Since the model finally works at the token-level, it is essential to reserve some training data to adapt the patch-level model to token-level. Specifically, we conduct patch-level training on a fraction $\lambda$ of the training data, and then use the resulting parameters to initialize the token-level model. Following this, the token-level model continues training on the remaining data to adapt the knowledge gained during patch-level training to the token-level. As illustrated in Figure 1, the overall training costs are reduced to $\lambda/K + 1 - \lambda$ times the original costs of token-level training. When the amount of training data is limited, this approach can also be utilized for efficient multi-epoch training. For example, given a budget of $N$ epochs, we can conduct patch-level training on the first $N\lambda$ epochs, and then switch to token-level training for $N(1 - \lambda)$ epochs.

## 3 EXPERIMENTS

### 3.1 SETUP

**Datasets.** We evaluate our approach on standard language modeling tasks, using the Pile dataset (Gao et al., 2020) containing 360B tokens for training [1]. We assess the performance of LLMs from multiple aspects, including their perplexity, zero-shot accuracy, and instruction-following ability. Perplexity is calculated on the WikiText-103 test set (Merity et al., 2017). We evaluate the zero-shot capabilities of language models on 6 NLP benchmarks, including MMLU (Hendrycks et al., 2021), HellaSwag (Zellers et al., 2019), PIQA (Bisk et al., 2020), WinoGrande (Sakaguchi et al., 2020), ARC-E, and ARC-C (Clark et al., 2018) [2]. For the pre-trained LLMs, we conduct instruction fine-tuning using the Alpaca dataset by GPT4 (Taori et al., 2023), and then evaluate their instruction-following abilities on MT-Bench (Zheng et al., 2024).

**Models.** We use the Transformer backbone (Vaswani et al., 2017) and adopt most of the architecture designs from LLaMA (Touvron et al., 2023a). We apply pre-normalization using RMSNorm (Zhang & Sennrich, 2019), use the SwiGLU activation function (Shazeer, 2020), and rotary positional embeddings (Su et al., 2021). We also apply FlashAttention-2 (Dao, 2024) to accelerate attention computation. We scale the model demension and obtain 4 different sizes of Transformers: Transformer-370M (hidden_size=1024, intermediate_size=2752, hidden_layers=24, attention_heads=16), Transformer-780M (hidden_size=1536, intermediate_size=4128, hidden_layers=24, attention_heads=16), Transformer-1.3B (hidden_size=2048, intermediate_size=5504, hidden_layers=24, attention_heads=16), Transformer-2.7B (hidden_size=2560, intermediate_size=6880, hidden_layers=32, attention_heads=32).

**Implementation Details.** Unless otherwise specified, the patch size $K$ is 4. The context length for token-level training 2048. For patch-level training, the context length is the patch size $K * 2048$. The global batch size is $2M$ tokens, and the total number of training steps is $N = 180000$. For patch-level training, the number of training steps is $N\lambda$, and then the model proceeds with token-level training for $N(1 - \lambda)$ steps. After patch-level training, only the obtained model parameters are used for initialization, and all other states like the optimizer and learning rate scheduler are reset. We use the tokenizer of LLaMA2, whose vocabulary size is 32000. Our models are optimized by the AdamW optimizer (Loshchilov & Hutter, 2019) with $\beta_1 = 0.9, \beta_2 = 0.95, \epsilon = 1e - 8$. The learning rate is $3e - 4$ and the cosine learning rate schedule is applied with warmup of 2000 steps. We use a weight decay of 0.1 and gradient clipping of 1.0, and no dropout is applied during training.

### 3.2 MAIN RESULTS

We train a series of LLMs of varying sizes (370M-2.7B parameters) on the Pile dataset. We employ patch-level training with the settings of $K = 4, \lambda = 2/3$, which theoretically reduces the training costs to $0.5\times$. Please refer to Appendix B for the actual speed measurement. For the Transformer-370M, we also explore other choices of $\lambda$ to evaluate its impact. Table 1 presents the performance comparison of the resulting models. Remarkably, our approach consumes only half of the compute

---

[1]Previous works generally refer to the Pile dataset as having 300B tokens, but our actual measurement is 360B. The discrepancy is likely due to differences in tokenizers; we use the LLaMA2 tokenizer, which has a relatively small vocabulary, possibly resulting in more tokens. The perplexity scores are also incomparable with models using other tokenizers.

[2]https://github.com/EleutherAI/lm-evaluation-harness

Table 1: Performance comparison of Transformers trained on the Pile dataset. $\lambda$ denotes the proportion of training data used for patch-level training, with the patch size $K$ fixed at 4. 'PPL' represents the perplexity score on the WikiText-103 test set. For zero-shot evaluations, we report the normalized accuracy across 6 NLP benchmarks. 'Average' means the average zero-shot accuracy.

| Model Type | Cost | PPL | MMLU | HellaSwag | PIQA | WinoG | ARC-E | ARC-C | Average |
|---|---|---|---|---|---|---|---|---|---|
| Transformer-370M | 1.0× | 10.9 | 22.9 | 40.8 | 67.5 | 53.1 | 44.3 | 24.7 | 42.2 |
| + Patch ($\lambda = 1/2$) | 0.625× | 10.6 | 23.5 | 42.0 | 67.9 | 52.1 | 46.1 | 25.6 | 42.9 |
| + Patch ($\lambda = 2/3$) | 0.5× | 10.7 | 23.7 | 41.1 | 68.0 | 51.9 | 46.0 | 24.2 | 42.5 |
| + Patch ($\lambda = 4/5$) | 0.4× | 11.0 | 23.3 | 40.5 | 67.5 | 51.7 | 44.9 | 24.5 | 42.1 |
| Transformer-780M | 1.0× | 9.2 | 24.4 | 48.5 | 69.0 | 55.4 | 49.0 | 26.7 | 45.5 |
| + Patch ($\lambda = 2/3$) | 0.5× | 9.1 | 24.1 | 49.1 | 70.6 | 54.8 | 51.3 | 28.2 | 46.3 |
| Transformer-1.3B | 1.0× | 8.2 | 23.9 | 54.5 | 71.2 | 57.3 | 55.1 | 28.9 | 48.5 |
| + Patch ($\lambda = 2/3$) | 0.5× | 8.2 | 24.3 | 54.1 | 71.6 | 57.8 | 55.6 | 30.4 | 49.0 |
| Transformer-2.7B | 1.0× | 7.1 | 25.3 | 62.2 | 74.3 | 61.5 | 61.2 | 34.3 | 53.1 |
| + Patch ($\lambda = 2/3$) | 0.5× | 7.2 | 25.4 | 61.9 | 74.9 | 62.4 | 61.9 | 34.6 | 53.5 |

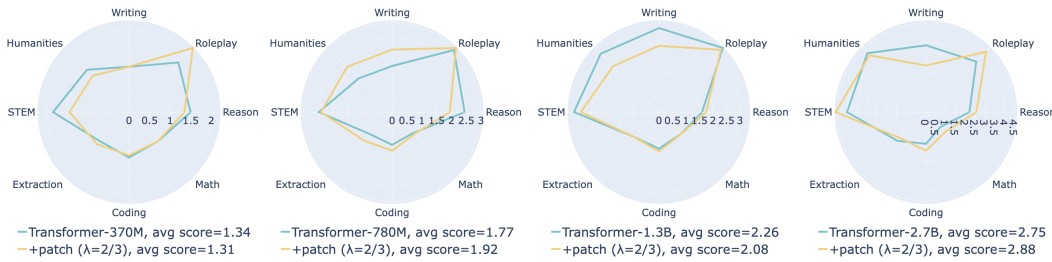

Figure 4: Instruction-following abilities evaluated on MT-bench, a multi-turn question set.

and incurs almost no performance loss. It matches the baseline model in terms of perplexity and even demonstrates a consistent gain in zero-shot evaluations, raising the average accuracy by approximately 0.5%. The model performance is also influenced by the choice of $\lambda$. Within the range of values we set, a smaller $\lambda$ leads to better model performance but also entails larger training costs. A more detailed study on the hyperparameter $\lambda$ will be presented in Section 3.6.

We further conduct instruction fine-tuning using the Alpaca dataset by GPT4 to examine the impact of patch-level training on the model's instruction-following ability. We evaluate our models using MT-Bench, a multi-turn question set, and present the experimental results in Figure 4. As can be seen, our approach maintains a similar instruction-following ability to the original models, with some experiencing a score decrease (Transformer-370M, Transformer-1.3B) and others showing an improvement (Transformer-780M, Transformer-2.7B), which can be viewed as regular variations.

Our primary motivation for patch-level training is to enhance the model's knowledge acquisition efficiency. Interestingly, experimental results show that this approach can sometimes lead to performance improvements, which is beyond our initial expectation. We initially thought that the extended context length in patch-level training contributes to the improvements. However, when we decreased the context length during patch-level training from $KT = 8192$ to $T = 2048$ for Transformer-370M ($\lambda = 1/2$), the model performance only experiences a slight decline (PPL $\uparrow 0.06$, zero-shot accuracy $\downarrow 0.2$), yet still surpasses the baseline, implying that context length is not the primary factor. We hypothesize that two other factors might be responsible for these improvements: firstly, the patch-level initialization could potentially serve as a form of regularization; secondly, by compressing consecutive tokens into patches, the model might more effectively recognize and capture long-range dependencies due to the reduced token distance.

Table 2: Performance comparison of Transformers trained on 60B tokens for 6 epochs.

| Model Type | Cost | PPL | MMLU | HellaSwag | PIQA | WinoG | ARC-E | ARC-C | Average |
|---|---|---|---|---|---|---|---|---|---|
| Transformer-370M | $1.0\times$ | 11.0 | 23.6 | 40.8 | 66.5 | 50.8 | 44.8 | 25.2 | 42.0 |
| + Patch ($\lambda = 1/2$) | $0.625\times$ | 10.4 | 23.9 | 43.3 | 67.5 | 55.6 | 44.4 | 26.1 | 43.5 |
| + Patch ($\lambda = 2/3$) | $0.5\times$ | 10.5 | 24.7 | 42.4 | 67.9 | 51.9 | 45.3 | 24.7 | 42.8 |
| + Patch ($\lambda = 4/5$) | $0.4\times$ | 10.7 | 23.0 | 41.5 | 67.0 | 52.0 | 45.1 | 25.4 | 42.3 |

## 3.3 MULTI-EPOCH TRAINING

Given that patch-level training consumes training data more rapidly, it is more data-hungry compared to token-level training. Consequently, it is essential to consider scenarios where training data is relatively limited and assess the performance of patch-level training when training data is reused for multi-epoch training (Muennighoff et al., 2023). We randomly extract a subset of 60B tokens from the Pile dataset and increase the number of training epochs to $N = 6$. In this way, the model is first trained on patch-level for $N\lambda$ epochs, followed by $N(1 - \lambda)$ epochs of token-level training.

The results presented in Table 2 demonstrate that patch-level training maintains its superiority in terms of training efficiency and performance on multi-epoch training. Intriguingly, when both consuming 360B tokens, patch-level training for multiple epochs even outperforms its single-epoch variant in Table 1. This unexpected advantage may stem from the synergistic effect of integrating patch-level and token-level training on the same data, which likely enhances model regularization. It also suggests that our approach can be data-efficient by initializing the model with patch-level training for one or multiple epochs, offering a promising direction for boosting model performance.

## 3.4 SCALING PROPERTIES

In the above, we have validated the effectiveness of patch-level training across several model sizes (370M-2.7B), using a training set of 360B tokens. However, state-of-the-art LLMs are generally trained on model sizes and datasets that are at least an order of magnitude larger than our settings. Therefore, it is crucial to know the scaling properties of patch-level training, i.e., how it performs when applied to larger training datasets and models.

In Table 1, we notice a trend related to the model size: the performance advantage of patch-level training appears to decrease as the model size increases. Table 3 describes this trend more precisely, indicating that the model with patch-level training experiences smaller performance gains from the increase in model size. On the other hand, Table 4 presents the changes of cross-entropy loss when maintaining a constant model size and varying the size of the training data. As the data size increases, the performance of patch-level training improves at a faster rate compared to the baseline model.

Table 3: Test losses when scaling the model size from 370M to 2.7B and training on the Pile dataset (360B tokens). '↓' indicates the loss reduction compared to the previous model size.

| Model Size | 370M | 780M | 1.3B | 2.7B |
|---|---|---|---|---|
| Transformer | 2.392 | 2.217 ($\downarrow$0.175) | 2.102 ($\downarrow$0.115) | 1.961 ($\downarrow$0.141) |
| + Patch ($\lambda = 2/3$) | 2.368 | 2.208 ($\downarrow$0.160) | 2.108 ($\downarrow$0.100) | 1.980 ($\downarrow$0.128) |

Table 4: Test losses of Transformer-370M when scaling the size of training data from 45B to 360B. '↓' indicates the loss reduction compared to the previous data size. The batch size is adjusted to maintain a consistent number of training steps.

| Data Size | 45B | 90B | 180B | 360B |
|---|---|---|---|---|
| Transformer | 2.526 | 2.460 ($\downarrow$0.066) | 2.423 ($\downarrow$0.037) | 2.392 ($\downarrow$0.031) |
| + Patch ($\lambda = 2/3$) | 2.553 | 2.468 ($\downarrow$0.085) | 2.413 ($\downarrow$0.055) | 2.368 ($\downarrow$0.045) |

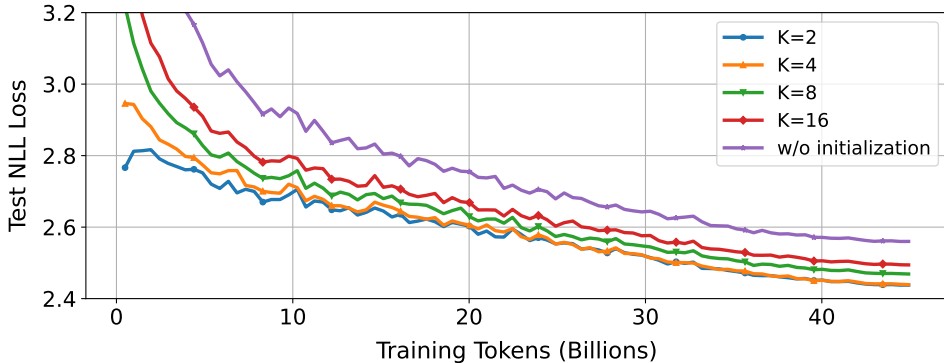

Figure 5: Test losses of Transformer-370M w.r.t the number of processed tokens. Models are initialized by patch-level training with patch size $K$.

This phenomenon can be explained from the perspective of knowledge transfer. As the data size increases, more training data is employed to adjust the model from patch-level to token-level, facilitating a smoother knowledge transfer process. However, an increase in model size implies a greater number of model parameters to be transferred to the token-level, which raises the level of transfer difficulty and necessitates more training data. Based on this explanation, patch-level training is better suited for scenarios with abundant training data.

Note that the above conclusions are drawn under the settings of $K = 4, \lambda = 2/3$, which may vary with changes in the patch size $K$ and the patch-level data fraction $\lambda$. At present, we have not identified a general scaling law for patch-level training that incorporates $K$ and $\lambda$. Instead, we have made some observations regarding their effects on model performance, which will be discussed in the following.

### 3.5 EFFECT OF PATCH SIZE $K$

We investigate the effect of patch size under the settings of 90B training tokens, 370M model parameters, a batch size of 512K, and $\lambda = 1/2$. The results are shown in Figure 5. Across different patch sizes, the loss curves for patch sizes $K = 2$ and $K = 4$ are nearly indistinguishable, while further increasing the patch size to 8 or 16 results in a certain performance decline. Despite this, these models still exhibit significant performance improvements when compared to the model trained from scratch, which does not benefit from the initialization of patch-level training.

Overall, the patch size of $K = 4$ strikes a favorable trade-off between training efficiency and performance. Considering that larger patch sizes can process more data at the same cost, we also experiment with patch-level training using $K = 8$ on 90B tokens, which costs similar compute as $K = 4$ on 45B tokens. Following this, both models proceed with token-level training on 45B tokens, and coincidentally, their loss curves are nearly identical. In this context, the advantage of $K = 4$ lies in its data efficiency, as it achieves similar performance while consuming less data.

### 3.6 EFFECT OF $\lambda$

The hyperparameter $\lambda$ allocates the ratio of training data between patch-level and token-level training. A larger $\lambda$ results in more tokens being compressed into patches, leading to a higher acceleration rate, but it may also leave insufficient data to adjust the model to the token-level. In this section, we investigate the effect of $\lambda$ under the settings of 370M model parameters, a batch size of 512K, and a patch size of $K = 4$. We consider two scenarios:

1. Unlimited computational budget: We assess the impact of varying $\lambda$ while keeping the data size constant (90B tokens). The results are shown in Figure 6.
2. Unlimited training data: We identify the optimal $\lambda$ under a fixed amount of computational budget (tokens + patches = 56.25B). For example, when $\lambda = 1/2$, the size of training data should be 90B tokens, with 45B tokens being compressed into 11.25B patches. The results are shown in Figure 7.

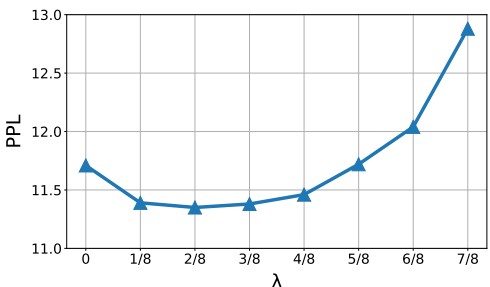 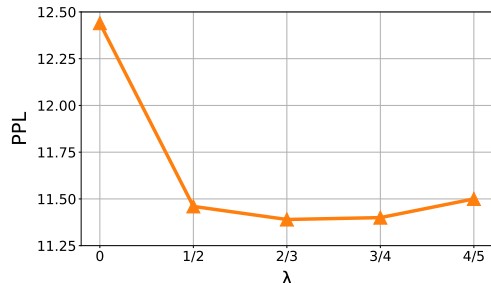

Figure 6: Effect of varying $\lambda$ while keeping the data size constant.

Figure 7: Effect of varying $\lambda$ while keeping the computational cost constant.

Figure 6 shows that the model performance initially rises and later falls as $\lambda$ increases, with a turning point near $\lambda = 1/4$. The performance improvements when $\lambda < 1/4$ can be attributed to the inherent benefits of patch-level training, as analyzed in Section 3.2. When $\lambda$ exceeds $3/4$, further increasing $\lambda$ leaves insufficient data to adjust the model to the token-level, leading to a rapid decline in performance. Figure 7, on the other hand, shows that when computational budget are limited, the optimal value for $\lambda$ is around $2/3$. Note that these conclusions are specific to the current settings and should be used as a reference only. The optimal $\lambda$ may vary depending on factors such as data size and patch size. To determine the optimal value of $\lambda$ in any scenario, it is essential to establish the scaling law for patch-level training.

## 3.7 EFFECT OF ARCHITECTURE

In the current setup, the architecture of the patch-level model is identical to that of the token-level model, facilitating smoother model transfer but also leading to some concerns. For instance, the token-to-patch transformation and the prediction of the next patch do not take into account the order of tokens within a patch. One may conjecture that patch-level models might benefit from adopting architectures that are better suited for representing and predicting patches, where the additional parameters introduced in such architectures could simply be discarded in the next stage.

We evaluate this strategy under the settings of 90B training tokens, 370M model parameters, a batch size of 512K, and $\lambda = 1/2$. Specifically, we incorporate a linear projection layer at both the input and output sides of the model. On the input side, the conversion of token embeddings into patch embeddings is facilitated through a linear projection $w_{in} \in \mathbb{R}^{Kd \times d}$. On the output side, a linear projection $w_{out} \in \mathbb{R}^{d \times Kd}$ is employed to transform patch-level representations back into token-level representations, followed by $K$ softmax layers to obtain the probability distribution of each token. The effects of these two modules are detailed in Table 5.

Table 5: Impact of architecture modifications in patch-level models. '+InputProj' and '+OutputProj' denote the incorporation of linear projections at model's input and output, respectively. 'Patch PPL' and 'Token PPL' are perplexities of the patch-level and token-level model, respectively.

| Model | Transformer | +InputProj | +OutputProj | +Both |
|---|---|---|---|---|
| Patch PPL | 159.17 | 146.93 | 99.48 | 86.50 |
| Token PPL | 11.46 | 11.63 | 11.50 | 12.33 |

Overall, while these modifications are effective in reducing the patch-level loss, they do not translate into benefits for the subsequent token-level training. Particularly, when linear projections are applied at both the model input and output, the performance of the subsequent token-level model significantly declines. It suggests that it is crucial for at least one end (input or output) of the patch-level model to remain consistent with the token-level model, which acts as an anchor to align their representations. It also shows that there is no direct correlation between patch-level loss and the final performance of the token-level model, so a large loss during patch-level training does not imply ineffective learning. Therefore, we opt to preserve the original Transformer architecture for patch-level training.

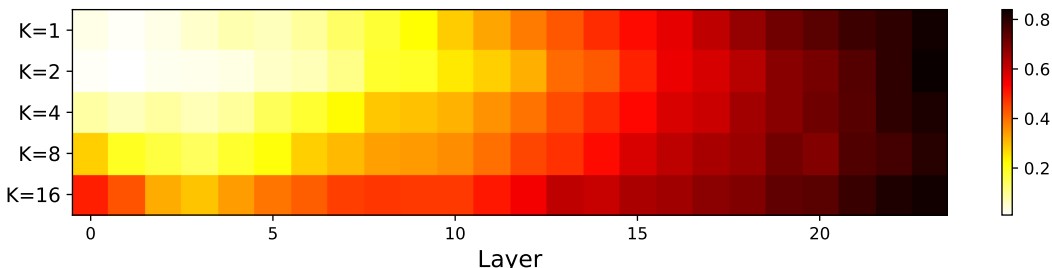

Figure 8: Percentage of activated neurons for models of different patch sizes. Output neurons of each model layer (FFN output) with an absolute value greater than 0.5 are classified as activated.

### 3.8 NEURON ACTIVATION

In this section, we quantitatively explain why patch-level training leads to better learning efficiency from the perspective of neuron activation. The training of LLMs is essentially a process of embedding knowledge from the training set into the model's parameters. During this process, the model employs all of its parameters to encode every token and updates the relevant parameters based on the gradient feedback. We argue that this is an inefficient process for large models, as the knowledge encapsulated in each token is only associated with a small subset of model parameters, resulting in a limited number of effectively activated and updated parameters.

We substantiate this by measuring the percentage of activated neurons for models of different patch sizes, as depicted in Figure 8. In the token-level model ($K = 1$), only a small proportion of neurons are activated, suggesting that the model has enough capacity to handle text units with higher information density, thus indicating significant room for improvement in training efficiency. By grouping multiple tokens into a patch, the information density processed at each step is increased, which is manifested as increased neuron activation rates. From this observation, it becomes evident that patch-level training makes more comprehensive utilization of the model's capabilities, therefore demonstrating higher training efficiency.

## 4 RELATED WORK

**Model Growth.** Our approach draws inspiration from transfer learning, reducing training costs by transferring knowledge acquired at a lower training cost (patch-level) to a model with a higher training cost (token-level). A similar strategy has been employed in studies of model growth, which train large models at a relatively lower cost by progressively increasing the model size during training. For example, Gong et al. (2019); Yang et al. (2020) improve the training efficiency by transferring knowledge from a shallow model to a deep model, where model layers are progressively stacked during training. Gu et al. (2021) further proposes progressive compound growth, where the model grows at multiple dimensions during training, including the context length, model width, and the number of layers. Subsequent studies primarily focus on the initialization problem during the model growth process, i.e., how to expand the small model into a large one. Chen et al. (2022); Yao et al. (2024) aim to achieve function-preserving growth (Chen et al., 2015) that the post-growth model have the same function as the pre-growth model, which intuitively ensures smooth knowledge transfer. Wang et al. (2023); Pan et al. (2023) introduce learnable linear operators that linearly map the parameters of the small model to initialize the large model. Compared to model growth, patch-level training is more flexible and generalizable as it does not necessitate specialized model architectures or carefully crafted model mapping strategies. More importantly, patch-level training merely alters the information density of text units while preserving the full set of model parameters. Therefore, it enables the model to develop its complete capabilities without compromising its performance.

**Multi-Token Prediction.** Our approach improves training efficiency by concurrently predicting all tokens in the next patch. Similar attempts of multi-token prediction have been made in the past to improve the inference efficiency, including non-autoregressive generation (Gu et al., 2018) and speculative decoding (Stern et al., 2018; Leviathan et al., 2023; Chen et al., 2023). Non-autoregressive

generation reduces the number of decoding iterations by generating all tokens at once, involving techniques such as knowledge distillation (Kim & Rush, 2016; Shao et al., 2022), training objectives (Shao et al., 2019; 2020; 2021; Ghazvininejad et al., 2020; Du et al., 2021; Shao & Feng, 2022; Ma et al., 2023; Shao et al., 2023), latent modeling (Kaiser et al., 2018; Ma et al., 2019; Shu et al., 2020), iterative decoding (Lee et al., 2018; Ghazvininejad et al., 2019; Gu et al., 2019), and expressive model architectures (Libovický & Helcl, 2018; Huang et al., 2022; Gui et al., 2023). Despite the reduction in decoding iterations, the computational demands (FLOPs) do not decrease and may even rise for certain structures, leading to increased training costs. Speculative decoding is a novel decoding paradigm for accelerating LLM inference. During each step of decoding, it drafts several future tokens efficiently and then verifies them in parallel. The model for generating draft tokens can either be a relatively small model (Leviathan et al., 2023; Chen et al., 2023) or a non-autoregressive model that generates multiple tokens in parallel (Cai et al., 2024; Lin et al., 2024; Fu et al., 2024; Li et al., 2024). Recently, Gloeckle et al. pointed out that besides accelerating the inference, multi-token prediction also serves as an auxiliary training task to enhance the training signal. The key difference between our approach and these works lies in our utilization of multi-token prediction for reducing sequence length during training, with the goal of improving training efficiency. Regarding the model structure, we avoid introducing extra parameters and employ a single head for multi-token prediction.

**Patch-Level Models.** The concept of handling input data at the patch-level has emerged as a pivotal strategy for enhancing computational efficiency and capturing local features. Convolutional Neural Networks (CNNs, Lecun et al., 1998) are perhaps the earliest attempt of patch-level processing, utilizing kernel filters to extract local features from images. More recently, Vision Transformers (Dosovitskiy et al., 2021) have revolutionized image processing by employing CNNs to encode an image into non-overlapping image patches, thereby enabling Transformers to efficiently capture both local and global features. Similarly, speech models also rely on CNNs to compress high-frequency waveforms into hidden state representations (Baevski et al., 2020; Hsu et al., 2021), which can be interpreted as speech patches. Dosovitskiy et al. (2021) suggested that the Vision Transformer should be fine-tuned at higher resolution, a strategy similar to ours, which improves data precision during the later phases of training. In subsequent studies, the impact of patch sizes has also garnered attention. Beyer et al. (2023) proposed to randomize the patch size at training time, which creates models capable of handling diverse patch sizes and adapting to different compute budgets. Further exploration by Anagnostidis et al. (2024) revealed that training with larger patch sizes is generally more computationally efficient. This insight led to a training strategy that progressively decreases the patch size, enabling more efficient training of vision Transformers. For textual data, characters, the basic building blocks of text, can be downsampled into more compact representations (Clark et al., 2022) or merged into tokens (Sennrich et al., 2016; Kudo & Richardson, 2018). Recently, there have been attempts to further compress tokens into patches, with the model directly processing the patch sequence (Nawrot et al., 2022; Mujika, 2023; Yu et al., 2024; Ho et al., 2024). However, it remains necessary to upsample the patch representation and input it into a token-level autoregressive model for the likelihood inference.

## 5 CONCLUSION

This paper introduces patch-level training, an efficient training approach for large language models, in which multiple tokens are aggregated into a unit of higher information density, referred to as a 'patch', to serve as the fundamental text unit for training LLMs. During patch-level training, the model reads training data in patches and learns to predict the next patch. Following this, a small amount of training data is utilized to adjust the model to the token-level. Experimental results show that this approach can cut LLM training costs by 50% while maintaining comparable performance.

Yet, our exploration of patch-level training is still in its infancy, and advancements in the following directions could further enhance this methodology: assessing the scalability of patch-level training by evaluating its performance on larger models and datasets; establishing an empirical scaling law for patch-level training, ideally incorporating both $K$ and $\lambda$; developing advanced training techniques to accommodate larger $K$ and $\lambda$, thereby pushing acceleration rates to a higher level; further investigating the potential of patch-level training in multi-epoch training; exploring the applicability of patch-level training to other data modalities, such as images, speech, and video.

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

## A PSEUDOCODE

```
# Model input
num_patches = seq_length // self.patch_size
inputs_embeds = inputs_embeds.view(batch_size, num_patches, self.
    patch_size, -1).mean(2)
position_ids = position_ids[:, :num_patches]

...

# Model output
shift_logits = logits[..., :-1, :].reshape(-1, self.config.vocab_size)
shift_labels = labels[..., self.patch_size:].reshape(-1, self.patch_size)
loss = 0
log_probs = F.log_softmax(shift_logits, dim=1)
for i in range(self.patch_size):
    loss = loss + F.nll_loss(log_probs, shift_labels[:, i])
loss = loss / self.patch_size
```

## B SPEED MEASUREMENT

The practical speed-up achieved through patch-level training falls short of the theoretical $K\times$ improvement, primarily due to overheads of data loading and gradient synchronization. Table 6 gives the actual running speed of patch-level training in comparison with the token-level baseline setting (patch_size=1, block_size=2048, per_device_train_batch_size=4, accumulation_steps=8), measured on 8 NVIDIA A100 GPUs. Increasing the patch size to 4 and the block length to 8192 results in a speed-up of approximately $3.8\times$, albeit with some efficiency loss due to data loading. In practice, the gradient accumulation steps should be reduced to 2 to keep the total batch size constant, which brings the speed-up down to around $3.5\times$ due to more frequent gradient synchronizations. Consequently, the actual runtime is approximately $\frac{2}{3} \times \frac{1}{3.5} + \frac{1}{3} \approx 0.523\times$, which is slightly larger than the theoretical cost of $0.5\times$.

Table 6: Speed comparison between token-level and patch-level training.

| Settings | Tokens / sec | Speed-up |
|---|---|---|
| Baseline | 247.1K | 1.0× |
| patch_size=4, block_size=8192 | 933.5K | 3.78× |
| patch_size=4, block_size=8192, accumulation_steps=2 | 864.2K | 3.50× |

## C MIXUP TRAINING

The core idea of this work is to aggregate multiple tokens into a unit of higher information density, which we refer to as a 'patch'. Our current strategy compresses every consecutive $K$ tokens into a single patch. Here we explore an alternative approach inspired by mixup data augmentation (Zhang et al., 2018). This method involves randomly selecting $K$ training samples and mixing their tokens position-wise to form a patch sequence. Similarly, we train the model with the next patch prediction objective.

We conduct experiments under the settings of 90B training tokens, 370M model parameters, a batch size of 512K, and $\lambda = 1/2$. The results are shown in Figure 9. Mixup training offers some improvements over the model trained from scratch; however, its performance significantly lags behind that of patch-level training. The primary difference between these methods lies in the composition of the patches, suggesting that it is more beneficial to form patches from consecutive tokens. We hypothesize that by compressing consecutive tokens into patches, the model may more effectively recognize and capture long-range dependencies, thanks to the reduced distance between tokens.

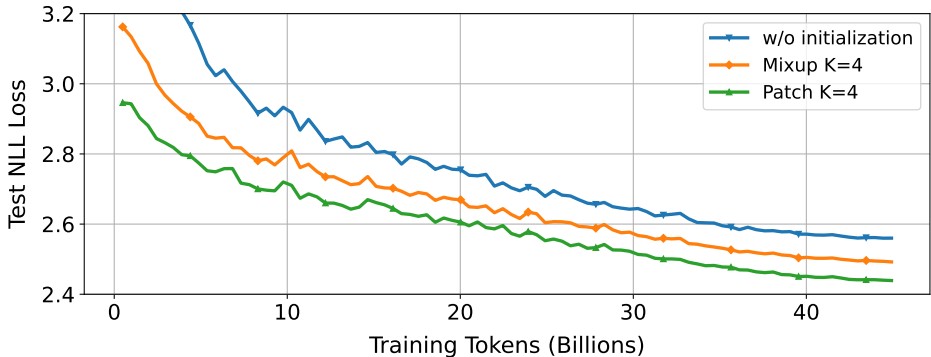

Figure 9: Test losses of Transformer-370M w.r.t the number of processed tokens. Models are initialized by mixup training or patch-level training with patch size $K$.

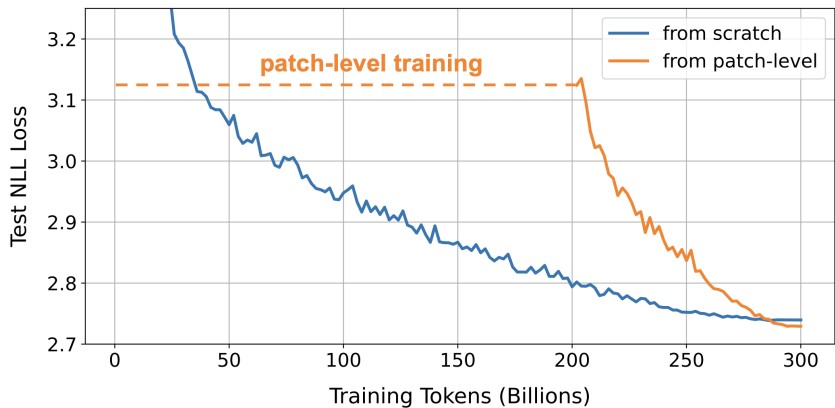

Figure 10: Negative log-likelihood (NLL) loss on test set w.r.t the number of processed tokens during the training of 370M-parameter Transformers. The LLaMA3 tokenizer is used.

## D  EFFECT OF TOKENIZER

Our approach enhances training efficiency by increasing the information density of text units and reducing their quantity. Another dimension that works similarly is the choice of tokenizer, as a tokenizer with a larger vocabulary typically represents the same data with fewer tokens. This raises an interesting question about how the efficacy of patch-level training is affected by the tokenizer. The LLaMA2 tokenizer, used in our initial experiments, has a vocabulary size of 32,000. Here, we extend our study to include the LLaMA3 tokenizer (Dubey et al., 2024), whose vocabulary size is 128,000, to observe whether our approach remains effective with tokenizers that have a higher compression rate.

We continue to use the Pile dataset for the training, which comprises approximately 300B tokens when encoded with the LLaMA3 tokenizer, corresponding to about 85% of the token count compared to the LLaMA2 tokenizer. Figure 10 depicts the loss curves for the models trained using patch-level training ($K = 4, \lambda = 2/3$) and from scratch at the token-level. Table 7 presents a performance comparison between the two approaches. Remarkably, patch-level training remains its effectiveness, achieving slightly better performance than token-level training while halving the training cost.

The choice of patch size, we hypothesize, is also swayed by the tokenizer's compression rate. When the token representations are less compact, a larger patch size might be employed to increase information density, and vice versa. We validate this hypothesis by comparing the effects of patch sizes $K = 2$ and $K = 4$ under the settings of 75B training tokens, 370M model parameters, a batch size of 512K, $\lambda = 1/2$, and using the LLaMA3 tokenizer. The results are illustrated in Figure 11.

Table 7: Performance comparison of Transformers trained on the Pile dataset. The LLaMA3 tokenizer is used.

| Model Type | Cost | PPL | MMLU | HellaSwag | PIQA | WinoG | ARC-E | ARC-C | Average |
|---|---|---|---|---|---|---|---|---|---|
| Transformer-370M | 1.0× | 15.48 | 22.9 | 41.2 | 67.3 | 54.1 | 45.6 | 25.8 | 42.8 |
| + Patch ($\lambda = 2/3$) | 0.5× | 15.32 | 23.1 | 41.1 | 68.1 | 53.2 | 45.9 | 26.2 | 42.9 |

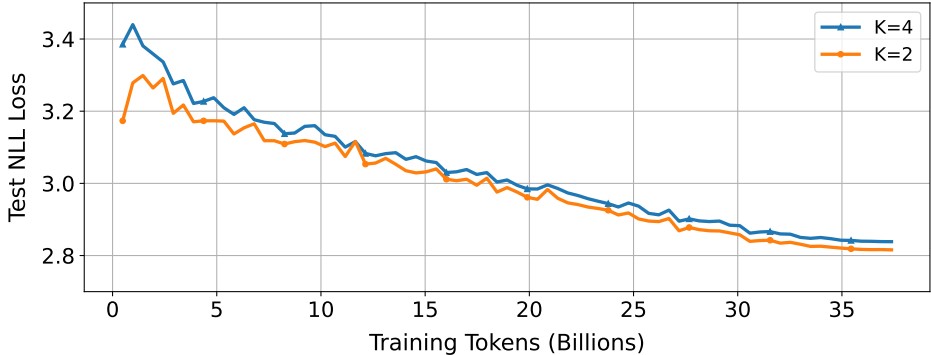

Figure 11: Test losses of Transformer-370M w.r.t the number of processed tokens. Models are initialized by patch-level training with patch size $K$. The LLaMA3 tokenizer is used.

It is observed that there is a slight performance gap between patch sizes $K = 2$ and $K = 4$ at this setting, whereas, in comparison, the loss curves for these patch sizes using the LLaMA2 tokenizer are almost indistinguishable in Figure 5. This confirms our hypothesis that the suitable patch size varies with the tokenizer. Nevertheless, we posit that a patch size of $K = 4$ is still more advantageous under this setting, offering significant efficiency improvements at the cost of a minor performance trade-off.

