# OpenReview forum: "Beyond Next Token Prediction: Patch-Level Training for Large Language Models"
_ICLR.cc/2025/Conference — ICLR 2025 Spotlight_

### Official Review · Reviewer_kTBD · 2024-10-31

**Soundness:** 4
**Presentation:** 3
**Contribution:** 4
**Rating:** 8
**Confidence:** 5

**Summary:**

This paper proposes a strategy to reduce training cost of LLMs using patch level training, where 'patch' refers to a sequence of consecutive tokens. The key idea is to perform training at the patch level initially whereby prior patches are used to predict the next patch, and then proceed with regular token level training.  The paper applies the approach to a range of model sizes (370M-2.5B) and shows that it can reduce the training cost by 50% without any performance degradation on model perplexity or downstream task performance. The paper presents variation in performance wrt patch size, model size, amount of training data and multi-epoch scenarios.

**Strengths:**

* Presents a simple approach to reduce training cost using a patch level training for initialization the regular token-level training
* Reports experiments on a range of model sizes showing that the approach can reduce training cost by 50% without any degradation in model performance

**Weaknesses:**

* There are some parts of the paper which are unclear. See questions below.
* It would be good to see some additional analysis/ablations to understand why the approach can lead to performance improvements.

**Questions:**

* L156: "Instead, we maintain a single output head and make its prediction cover all tokens in the next patch". How is this done? Does your output vocabulary have a size of K*|V| where |V| is the number of symbols ?
* Equation 3 uses 'p' to refer to both probabilities and patches. Consider using different symbols for the two.
* Sec 3.7: Have you run experiments adding more transformer layers for mapping the token embeddings to patch embeddings and vice-versa at the output side? It is possible that a linear mapping is not powerful enough.

---

> ### Author Response · Authors · 2024-11-19
> **Author response**
>
> > Additional analysis/ablations to understand why the approach can lead to performance improvements.
>
> We appreciate the reviewer's valuable suggestion. We have recently explored an alternative training strategy, mixup training, which provides some insights on the performance gains. In mixup training, we randomly select K training samples and mix their tokens position-wise to form a patch sequence, and similarly train the model with next patch prediction. We share the experiment results in Appendix C. Despite the methodological similarities, the performance of mixup training significantly lags behind that of patch-level training. The primary difference between these methods lies in the composition of the patches, suggesting that it is more beneficial to form patches from consecutive tokens. Therefore, we now lean towards the following interpretation of the performance gains: by compressing consecutive tokens into patches, the model may more effectively recognize and capture long-range dependencies, thanks to the reduced distance between tokens.
>
>
> > Does your output vocabulary have a size of K*|V| where |V| is the number of symbols ?
>
> The size of output vocabulary remains |V|. Based on the predicted log-probabilities on the vocabulary, we compute the cross-entropy loss against all tokens in the next patch. In this way, we treat the patch prediction as a mixture of token probabilities within that patch, effectively capturing the patch's overall likelihood without expanding the vocabulary size.
>
> To improve clarity on our methodology, we have updated our manuscript to include the below pseudocode in Appendix A, which outlines the process for patch representation and prediction:
>
> ```
> # Model input
> num_patches = seq_length // self.patch_size
> inputs_embeds = inputs_embeds.view(batch_size, num_patches, self.patch_size, -1).mean(2)
> position_ids = position_ids[:, :num_patches]
> ...
> # Model output
> shift_logits = logits[..., :-1, :].reshape(-1, self.config.vocab_size)
> shift_labels = labels[..., self.patch_size:].reshape(-1, self.patch_size)
> loss = 0
> log_probs = F.log_softmax(shift_logits, dim=1)
> for i in range(self.patch_size):
>     loss = loss + F.nll_loss(log_probs, shift_labels[:, i])
> loss = loss / self.patch_size
> ```
>
> > Equation 3 uses 'p' to refer to both probabilities and patches. Consider using different symbols for the two.
>
> We thank the reviewer for raising this concern. To avoid the confusion, we have revised the manuscript to use uppercase $P$ to denote probabilities.
>
> > Adding more transformer layers for mapping the token embeddings to patch embeddings and vice-versa at the output side.
>
> We have not yet conducted experiments with adding more transformer layers. Our intuition suggests that it may cause substantial challenges to align the representation spaces of the two models, potentially complicating the transition from patch-level to token-level representations. We posit that the task of next patch prediction already equips the model with a sufficient understanding of the data distribution. What remains less developed at this stage is the model's grasp of the internal ordering of words within patches, a skill that we believe can be readily acquired during the subsequent token-level training.

---

> > ### Comment · Reviewer_kTBD · 2024-11-21
> >
> > Thanks for your responses. They answer my questions.

---

### Official Review · Reviewer_o6fE · 2024-10-31

**Soundness:** 3
**Presentation:** 4
**Contribution:** 3
**Rating:** 8
**Confidence:** 4

**Summary:**

The paper presents an interesting approach to accelerate the costly pre-training of language models. The core idea is to change the basic unit from a token to patch which comprises several tokens. This results in computational savings for the main transformer model (the input encoding and the model prediction still produces tokens but it does so based on a patch encoding).

**Strengths:**

The paper presents an interesting approach to accelerate language model training, specifically:
* the approach appears to be novel
* the paper is well written and clear in the presentation
* the approach is thoroughly evaluated (different (small) model sizes, including an array of ablations, an alternative approach to computing patches etc.)

**Weaknesses:**

* The evaluation is on the smaller scale of language mode sizes. Evaluation for larger model sizes is very challenging but it would be a large plus to understand the scalability of the method.
* Table 3/4 show PPL results and draw conclusions based on the difference between PPLs in the range of ~10 and later on in ~7. This assumes that PPL is a linear measure but it is in fact an exponential quantity based on cross entropy. Therefore, the actual entropy difference between PPL 11 and PPL 10 is far lower than it is between, say PPL 8 and PPL 7. I would encourage the authors to simply report entropy instead of PPL for this particular analysis to make this clearer.

**Questions:**

* Line 266 mentions that the context length in the patch level training has little effect. Using context length KT performed as well as just T. However, these conclusions were drawn based on a very small model with 370M parameters which may not be able to exploit the larger context as well as higher capacity models. Did you perform a comparison on a larger model?
* Does the patch level training objective have any notion of which token position is currently being predicted within a patch (eq 3)? It appears to treat a patch as a bag of words with no order? This would be interesting to contrast with a setting where there is a notion of order.
* Section 3.5 analyses the patch size K and deems K=2/4 to have a good speed/accuracy tradeoff in your setting which is a two-stage procedure: (1) train with some K>1 and then (2) use K=1 at the end of training. However, you could also consider a more general setting where there are many more stages and where where you start with very large K and then progressively decrease K until you end up with K=1. Have you considered this?

---

> ### Author Response · Authors · 2024-11-19
> **Author response**
>
> > Evaluation for larger model sizes is very challenging but it would be a large plus to understand the scalability of the method.
>
> We agree with the reviewer's perspective that the scale of current evaluation remains insufficient. Further scaling up the experiments poses a significant challenge due to the substantial computational resources required, which we currently lack. As an alternative, we have explored the scaling properties of patch-level training in Section 3.4, which could estimate its scaling behavior to some extent.
>
> We hypothesize that scaling up patch-level training could exhibit more compelling properties. For instance, when there is an abundance of training data, a smaller fraction of the training data might suffice to adapt the model to token-level. Similarly, with an increase in model capacity, it could potentially accommodate a higher information density with an enlarged patch size $K$. We hope future work could explore the characteristics of patch-level training at scale, unlocking higher training efficiency to promote the advancement of next-generation LLMs.
>
> > Report cross-entropy instead of PPL.
>
> We appreciate the reviewer's valuable suggestion. In response, we have revised the manuscript to report the cross-entropy loss in Table 3/4. This change does not alter the conclusions drawn from our results, as our primary comparisons are made within the same column. These PPLs are within a similar range, and thus, the variations in cross-entropy basically align with the changes in PPL.
>
> > Line 266 mentions that the context length in the patch level training has little effect. Did you perform a comparison on a larger model?
>
> We have not conducted experiments on larger models. However, in an early experiment with a smaller model (~200M parameters, trained on 90B tokens, $\lambda=1/2$), we have observed a similar outcome. When employing a context length of KT for patch-level training, we noted a marginal 0.07 decrease in perplexity compared to using a context length of T. This also suggests that while there is a slight benefit to increasing context length, it does not fully account for the improvements observed in patch-level training.
>
> > Does patch level training treat a patch as a bag of words with no order? This would be interesting to contrast with a setting where there is a notion of order.
>
> The reviewer is correct that a patch is treated as a bag of words with no order. In Section 3.7, we have also experimented with other model architectures that incorporate token positions into patch prediction. Specifically, a linear projection $w_{out} \in \mathbb{R}^{d \times Kd}$ is employed to transform the patch representation into token representations, followed by a softmax layer to predict the probabilities of each token. The results show that, although incorporating positional information can reduce the loss during patch-level training, it does not translate into performance improvements in the subsequent token-level training. Furthermore, the token-level performance significantly drop when the positional information is incorporated at both the model input and output. We posit that it is crucial for at least one end (input or output) of the patch-level model to remain consistent with the token-level model, which acts as an anchor to align their representations.
>
> > A more general setting where there are many more stages and where you start with very large K and then progressively decrease K until you end up with K=1.
>
> We have not yet explored the approach of annealing the patch size during training, but it indeed sounds like a viable strategy, allowing the model to gradually transition to token-level over time. Future work could potentially investigate such advanced training techniques to pursue further acceleration and performance improvements.

---

> > ### Comment · Reviewer_o6fE · 2024-11-22
> > **Author response**
> >
> > I appreciate the clarifications and thank the authors for taking the time to craft a response.

---

### Official Review · Reviewer_zf1q · 2024-11-02

**Soundness:** 3
**Presentation:** 3
**Contribution:** 3
**Rating:** 8
**Confidence:** 4

**Summary:**

This work introduces a novel pretraining scheme aimed at reducing computational cost of pretraining. The core idea is to reduce the effective length of training sequences by concatenating and averaging the token representation of mutliple tokens into a patch representation. Then the model predicts tokens from the next patch using the output representation of the next patch. The simplicity of this idea is very intriguing as it does not require any extra parameters in the model. After such path-level pretraining with higher efficiency authors go back to tokenlevel training in order to ensure that the model will adjust to next token prediction representation.

Authors completed experiments showing that their method scales well with larger models and supports both streaming and multi-epoch training scenario. Extensive benchmarking showed that models trained with their approach are cheaper to train and are better or on the same level of performance as baseline models.

**Strengths:**

1. the method is easy to implement and brings original contribution, i see it being used a lot in the community and industry if it will pass the proof of concept in real world scenario.
2. High quality experiments cover wide range of model sizes and data settings. In addition, wide range of benchmarks are covered to evaluate the final model performance compared with usual token level training.
3. Assuming that this approach would scale with even larger models, this work provides a significant contribution in the pretraining research.

**Weaknesses:**

* Unclear conclusions about the effect of architecture. In the architecture ablation the observations are suggesting that there are might be some use cases where subsequent token level training does not work very well. IMO this might be important to highlight better in the text to avoid overselling the method.
* No numbers showing the actual speed improvements. Since all token projections are computed in patch level training, the expensive softmax operations over vocabularies have to be done. So it is interesting what is the speed improvement factor as we change K (patch size) in practice on real hardware.

**Questions:**

* Following my last weakness point, did you measure the speed up coming from patch level training in practice? e.g. in tokens per second metric or so. This would be very interesting to know.

---

> ### Author Response · Authors · 2024-11-19
> **Author response**
>
> > Unclear conclusions about the effect of architecture. Highlight some use cases where subsequent token level training does not work very well.
>
> We thank the reviewer for this valuable suggestion. In the architecture ablation, we observed that it is crucial for at least one end (input or output) of the patch-level model to remain consistent with the token-level model, which acts as an anchor to align their representations. Overall, the best practice is to keep the model architecture unchanged, as it facilitates smooth knowledge transfer between the two training phases. We have updated the manuscript to emphasize these concepts in Section 2 and Section 3.7.
>
> > No numbers showing the actual speed improvements.
>
> We thank the reviewer for raising this concern. First, we would like to address a misunderstanding about the computational process of patch prediction, which actually only needs one softmax operation to predict the next patch. Below is the key modifications in our code. On the input side, we average the token embeddings to obtain the patch embedding, with a negligible complexity of $\mathcal{O}(KTd)$. On the output side, we first perform a softmax operation to predict the log-probability of next patch, and then calculate cross-entropy losses for each token in the next patch. The added computational burden for this loss computation remains modest at $\mathcal{O}(KT)$.
> ```python
> # Model input
> num_patches = seq_length // self.patch_size
> inputs_embeds = inputs_embeds.view(batch_size, num_patches, self.patch_size, -1).mean(2)
> position_ids = position_ids[:, :num_patches]
> ...
> # Model output
> shift_logits = logits[..., :-1, :].reshape(-1, self.config.vocab_size)
> shift_labels = labels[..., self.patch_size:].reshape(-1, self.patch_size)
> loss = 0
> log_probs = F.log_softmax(shift_logits, dim=1)
> for i in range(self.patch_size):
>     loss = loss + F.nll_loss(log_probs, shift_labels[:, i])
> loss = loss / self.patch_size
> ```
>
> The reviewer is correct that the practical speed-up achieved through patch-level training falls short of the theoretical $K\times$ improvement, primarily due to overheads of data loading and gradient synchronization. Compared with the basic setting (patch_size=1, block_size=2048, per_device_train_batch_size=4, gradient_accumulation_steps=8, num_gpu=8), increasing the patch size to 4 and the block length to 8192 results in a speed-up of approximately 3.8$\times$, with some efficiency loss due to data loading. In practice, the gradient accumulation steps should be reduced to 2 to keep the total batch size constant, which brings the speed-up down to around 3.5$\times$ due to more frequent gradient synchronizations. Consequently, the actual runtime is approximately $\frac{2}{3} \times \frac{1}{3.5} + \frac{1}{3} \approx 0.523\times$, which slightly exceeds the theoretical cost of $0.5\times$. We have revised our manuscript to include a detailed analysis of the practical speed improvements in Appendix B.

---

> > ### Comment · Reviewer_zf1q · 2024-11-27
> >
> > Thank you for your clarifications and explanations!

---

### Official Review · Reviewer_QqZw · 2024-11-04

**Soundness:** 3
**Presentation:** 3
**Contribution:** 3
**Rating:** 6
**Confidence:** 3

**Summary:**

This paper seeks to reduce the training cost of LLMs by using patch-level training in the initial stage. A patch consists of multiple consecutive tokens, and patch-level representation can reduce training costs due to shorter sequence representations. Experiments performed across four model sizes and six NLP tasks show that the proposed method can reduce training costs by up to 50% with little to no performance degradation.

**Strengths:**

1. Reducing LLM training costs is an important research question. The problem formulation, motivation, and improvement objective of the paper are clear.

2. The proposed method is described clearly, and there seem to be adequate details to reproduce the work. The cost reduction of the proposed training method is significant and robust across multiple LLM sizes and NLP tasks.

3. The experiments cover a wide range of settings such as the fraction of patch-level training (lambda), patch size (K), model size, etc.

**Weaknesses:**

1. The optimal hyperparameters and intuitions derived from the work may be very specific to the settings associated with the paper. For example, the optimal values of lambda and patch size may be sensitive to the tokenizer. Therefore, the improvements of the proposed work may not generalize similarly to a new tokenizer.

2. It is not obvious whether the patch-level pretraining is suitable for larger scales. There are issues with training stability and convergence that only occur when the model size is scaled to several billions.

**Questions:**

N/A

---

> ### Author Response · Authors · 2024-11-19
> **Author response**
>
> > The optimal hyperparameters are specific to the settings. The optimal values of $\lambda$ and patch size may be sensitive to the tokenizer.
>
> We fully agree with the reviewer that the optimal hyperparameters may vary across different settings. As we have stated in Section 3.6, the conclusions are specific to the current settings and should be used as a reference only. Users are encouraged to adjust these hyperparameters based on their specific scenarios. For instance, when there is an abundance of training data, we can consider setting $\lambda$ slightly above 2/3 to achieve a higher acceleration rate. Similarly, for larger model sizes, there could be sufficient model capacity to accommodate higher information density with a larger patch size K. The perspective on tokenizers mentioned by the reviewer is particularly intriguing, as the choice of tokenizer also affects the data compression rate. A smaller vocabulary in the tokenizer typically results in more tokens for the same data, in which case a larger patch size might be employed to increase information density, and vice versa.
>
> Our experiments have primarily utilized the LLaMA2 tokenizer with a vocabulary size of 32,000, which is relatively small compared to other commonly used tokenizers in the industry. In light of the reviewer's suggestion, we have initiated experiments using the LLaMA3 tokenizer, which has a vocabulary size of 128,000 and produces approximately 85% of the token count generated by the LLaMA2 tokenizer. We plan to test the original settings ($\lambda=2/3$, patch size $K=4$) with this new tokenizer to observe the impact of vocabulary size on our method's effectiveness. We will also compare the performance of different patch sizes under this new tokenizer. The experiments are ongoing, and the results will be uploaded before the end of the discussion period. Please stay tuned for further updates.
>
> > It is not obvious whether the patch-level pretraining is suitable for larger scales.
>
> We acknowledge the risks associated with scaling, including issues related to training stability and convergence that were not encountered in our current experiments. Without direct experimentation, we cannot guarantee the absence of such issues. However, we posit that the risks associated with scaling patch-level training are relatively minor, owing to its inherent simplicity. Our approach retains the core mechanics of language model training and merely involves utilizing text units of a different granularity. This minimal alteration suggests that patch-level training could offer a scalable, efficient alternative without significantly altering the training dynamics or model architecture.

---

> ### Author Response · Authors · 2024-11-25
> **Effect of Tokenizer**
>
> Dear Reviewer,
>
> Apologies for the delay in our response as we have just completed the necessary experiments. We greatly appreciate your patience and understanding during this period. Please refer to Appendix D for our analysis on the effect of tokenizer. In short, we observe that patch-level training remains its effectiveness when using the LLaMA3 tokenizer. While halving the training costs, the performance of patch-level training is slightly better than token-level training:
>
> | Model Type                  | Cost      | PPL   | MMLU | HellaSwag | PIQA | WinoG | ARC-E | ARC-C | Average |
> |-----------------------------|-----------|-------|------|-----------|------|-------|-------|-------|---------|
> | Transformer-370M            | 1.0×      | 15.48 | 22.9 | 41.2      | 67.3 | 54.1  | 45.6  | 25.8  | 42.8    |
> | + Patch (${\lambda=2/3}$)   | 0.5×      | 15.32 | 23.1 | 41.1      | 68.1 | 53.2  | 45.9  | 26.2  | 42.9    |
>
>
> We also notice that the suitable patch size varies with the tokenizer used. It is observed that there is a slight performance gap between patch sizes $K=2$ and $K=4$ when using the LLaMA3 tokenizer, whereas, in comparison, the loss curves for these patch sizes using the LLaMA2 tokenizer are almost indistinguishable. Please refer to Appendix D for a comprehensive description.
>
> As the discussion period approaches its end (November 26th), we would like to ensure that our response has effectively addressed your concerns (we would also greatly appreciate if you could increase your score as you see fit to reflect this). Please let us know if you have any additional questions and we will be more than happy to address them. Thank you and look forward to hearing from you.
>
> Best regards,
> The Authors

---

> ### Author Response · Authors · 2024-12-02
> **Reminder**
>
> Dear Reviewer, we would like to kindly remind you that the discussion period is closing soon. Please let us know if you have any additional questions and we will be more than happy to address them.

---

### Author Response · Authors · 2024-11-19
**General response**

We sincerely thank all the reviewers for their great efforts in reviewing our work. In response to the valuable feedback, we have made several major updates to our manuscript, as outlined below:

1. In Appendix A, we provide the pseudocode for patch-level training to enhance its clarity.
2. In Appendix B, we give a detailed analysis of the practical speed improvements through patch-level training.
3. In Appendix C, we introduce an alternative mixup training strategy that we have recently explored. This section also includes a comparative analysis of its performance against patch-level training.
4. In Appendix D, we analyze the effect of tokenizer on patch-level training.

---

### Comment · Area_Chair_zKdq · 2024-11-21
**Please respond and update the score if necessary**

Dear Reviewers,

Kindly ensure that you respond proactively to the authors' replies so we can foster a productive discussion. If necessary, please update your score accordingly. We greatly appreciate the time and effort you’ve dedicated to the review process, and your contributions are key to making this process run smoothly.

Thank you,

AC

---

### Meta-Review · Area_Chair_zKdq · 2024-12-21

**Metareview:**

The paper introduces a pre-training method designed to reduce the training costs of large language models by implementing patch-level training, where multiple consecutive tokens are aggregated into a single patch. This approach can decrease computational expenses by up to 50% while maintaining performance levels. Experiments show the method scales effectively with larger models and supports both streaming and multi-epoch training scenarios, providing cost-effective solutions without additional model parameters. The models trained using this technique perform as well as or better than traditional baseline models.

The paper's method effectively reduces LLM training costs without compromising performance. It is thoroughly evaluated across a range of model sizes, settings, and benchmarks, showcasing robust performance. The approach is straightforward to implement and could achieve widespread adoption in both research and industry, making a substantial contribution to the field of pretraining research. The paper could benefit from additional analysis involving larger models.

Additionally, the authors have addressed most of the reviewers' questions thoughtfully and thoroughly. I recommend accepting this paper, and I am excited to see it presented at the conference.

**Additional Comments On Reviewer Discussion:**

All reviewers, except Reviewer QqZw, acknowledged the author response. Reviewer QqZw criticized the choice of hyper-parameters and noted a lack of clarity regarding scaling the experiments to larger model sizes, as the analysis was limited to 370 million parameters.

---

### Decision · Program_Chairs · 2025-01-22

Accept (Spotlight)